# Effect of Nanoparticle Size on the Mechanical Strength of Ni–Graphene Composites

**DOI:** 10.3390/ma14113087

**Published:** 2021-06-04

**Authors:** Karina A. Krylova, Liliya R. Safina, Ramil T. Murzaev, Julia A. Baimova, Radik R. Mulyukov

**Affiliations:** 1Institute for Metals Superplasticity Problems of the Russian Academy of Sciences, Khalturina 39, 450001 Ufa, Russia; murzaevrt@gmail.com (R.T.M.); julia.a.baimova@gmail.com (J.A.B.); radik@imsp.ru (R.R.M.); 2Department of Physics and Technology of Nanomaterials, Bashkir State University, Validy Str. 32, 450076 Ufa, Russia; 3Department of Metal Technology in Oil and Gas Engineering, Ufa State Petroleum Technological University, Kosmonavtov Str. 1, 450062 Ufa, Russia; saflia@mail.ru

**Keywords:** crumpled graphene, Ni–graphene composite, molecular dynamics, mechanical properties

## Abstract

The effect of the size of nickel nanoparticles on the fabrication of a Ni–graphene composite by hydrostatic pressure at 0 K followed by annealing at 1000 and 2000 K is studied by molecular dynamics simulation. Crumpled graphene, consisting of crumpled graphene flakes interconnected by van der Waals forces is chosen as the matrix for the composite and filled with nickel nanoparticles composed of 21 and 47 atoms. It is found that the main factors that affect composite fabrication are nanoparticle size, the orientation of the structural units, and temperature of the fabrication process. The best stress–strain behavior is achieved for the Ni/graphene composite with Ni${}_{47}$ nanoparticle after annealing at 2000 K. However, all of the composites obtained had strength property anisotropy due to the inhomogeneous distribution of pores in the material volume.

## 1. Introduction

Graphene is a new two-dimensional (2D) structure [1] that is already well-known by its unique physical and mechanical properties, such as its Young’s modulus at about 1 TPa [2], high strength at ∼130 GPa [2,3], high thermal conductivity at ∼5000 W m${}^{-1}$ K${}^{-1}$, and high mobility of electron carriers at ∼15,000 cm${}^{2}$ V${}^{-1}$ s${}^{-1}$ at room temperature [4,5], to name a few. Due to its unique properties, the idea of obtaining composite materials consisting of a metal matrix and graphene has been developed for the past decades. It was found that the introduction of graphene into a metal matrix results in an increase its mechanical resistance and strength [6,7,8,9,10]. Graphite inclusions improve the impact strength of layered carbon/Ni and carbon/Cu composites [11]. Even a small addition of graphene to a nickel matrix can increase its tensile strength and plastic elongation to 25% and 36%, respectively, compared to pure nickel [12].

The review by [13] contains a large number of experimental works on the fabrication and study of the metal matrix (including nickel matrix) graphene-reinforced composites. These composite properties are presented, and the factors influencing these properties are explained. For example, the graphene agglomerate formation negatively affects the composites’ mechanical properties, and the coherent graphene–matrix interface formation leads to an increase in the mechanical properties. It was also shown that the production method of a composite has a strong influence on its properties. Table 1 presents a summary of the commonly used methods for obtaining Ni/graphene composites, with an indication of the final mechanical properties. It can be seen that the microhardness (HV), ultimate tensile strength ($\sigma_{UTS}$), and yield stress ($\sigma_{0.2}$) differ significantly depending on the method used to obtain this composite based on a nickel matrix.

**Table 1 materials-14-03087-t001:** Production methods and varying mechanical properties of Ni/graphene composites: HV—microhardness, $\sigma_{0.2}$—yield stress, $\sigma_{UTS}$—ultimate tensile strength, and *E*—Young’s modulus.

Ref.	Materials	Production Method	Mechanical Properties
[10]	Graphene-enabled	Compressed and sintered	$\sigma_{0.2}$ = 780 MPa
	Ni/Ni${}_{3}$C composite	at 1723 K	$\sigma_{UTS}$ = 1095 MPa
			*E* = 222 GPa
[14]	(1) Graphene thickness	Electrochemical deposition	HV = 4.6 GPa
	3–5 nm		*E* = 240 GPa
	(2) Graphene thickness	Casting and fraction stirring	HV = 66 kg mm${}^{-2}$
	10–20 nm (1.2 vol.%)		
	(3) Graphene thickness	Semi-powder metallurgy	$\sigma_{UTS}$ = 197 MPa
	5–15 nm (0.3 wt.%)		
	(4) Graphene thickness	Hot extrusion	$\sigma_{UTS}$ = 238 MPa
	5–15 nm (0.3 wt.%)		
[15]	Graphene/Ni composite	Pulse-reverse electrodeposition	HV = 1036.9 MPa
	1.2 nm thick		*E* = 185 GPa
[16]	Graphene/Ni powders	(1) Cold pressing and	HV = 1.56 GPa
		annealing at 1523 K	
		(2) Cold pressing, annealing and	$\sigma_{0.2}$ = 1048 MPa
		high pressure torsion at 296 K	$\sigma_{UTS}$ = 1201 MPa
		(3) Cold pressing, annealing and	$\sigma_{0.2}$ = 923 MPa
		high pressure torsion at 473 K	$\sigma_{UTS}$ = 992 MPa
[17]	Graphene nanoplatelets/Ni	Powder metallurgy route	HV = 1.65 GPa
	nanocomposite powders	followed by the spark plasma	$\sigma_{0.2}$ = 370 MPa
		sintering process at 1073 K	

Layered metal/carbon composites were also widely studied. In [18], it was shown that a decrease in metal layer thickness in layered graphene/FCC metal composites results in strengthening of the material. The presence of graphene layers also determines the deformation mechanisms of the composite. In [19], it was shown that the Ni/graphene composite coating has excellent tribological properties due to the added graphene, which forms a lubricating film, which effectively reduces the friction coefficient and increases wear resistance. Excellent lubricating properties are demonstrated by a Cu-based nanocomposite containing graphene nanoflakes [20]. In addition, the electro-co-deposition of Ni/graphene oxide composite coating on low carbon steel was an effective anti-corrosion coating [21]. The Ni/graphene composite synthesized by electrodeposition on the nickel matrix surface showed a significant increase in mechanical properties [22].

The mechanical properties of layered graphene metal composites are strongly influenced by the graphene/metal interface. On the one hand, it can prevent the dislocations gliding, which reduces the strength properties of the composite. On the other hand, such interfaces can be a dislocation source in the metal matrix, leading to a degradation in mechanical properties or even failure [23,24,25,26,27,28,29]. However, there are composites based on a graphene matrix filled with metal nanoparticles, in which the metal/graphene interfaces are blurred. Metal nanoparticles especially are of great interest nowadays [30,31,32]. The basis for such structures can be crumpled graphene (CG), which consists of graphene nanoflakes connected by van der Waals forces [33,34]. Crumpled graphene is a new and promising structure with a high specific surface area of about 3523 m${}^{2}$/g. Such a three-dimensional (3D) architecture exhibits advantages from its particular structure: pores and cavities for serving metal nanoparticles, rigid bones for better strength, good interconnection between metal particles and graphene nanoflakes. Crumpled graphene shows considerable non-linearity in strain hardening under applied strain [34,35,36,37,38]. The difference between loading and unloading stress–strain curves reveals that this is completely non-elastic media. The important advantage that such a morphology of the crumpled graphene structure capable of absorbing large amounts of other atoms in its cavities, which was shown in our previous works [39,40].

Experimental production of carbon metal/graphene composites is a rather expensive and energy-consuming process. Since the structure peculiarities considerably depend on the experimental technique applied to obtain metal/graphene composite [41,42,43], different simulation methods can be very helpful for a basic understanding of physical and mechanical properties. Therefore, theoretical scientific research based on molecular dynamics (MD) simulation is widely used. The simulation allows us to analyze at the atomic level the dislocations evolution at the Cu/graphene interface [24,44], to estimate the tensile capacity of carbon nanotube/Al composites [45], to study the effect of aluminum orientation on the strengthening mechanisms of graphene/Al composites [46], and to predict the existence of metal/graphene composites based on crumpled graphene [39,47] and much more. Using MD simulation, it was also found that the addition of carbon materials (such as graphene or carbon nanotubes) to a metal matrix leads to its strengthening [26,48,49,50,51].

In most theoretical [24,26,41,42,43,44,45,46,48,49,50,51] and experimental works [6,7,8,9,11,12,23,25,27,28,29], metal–matrix composites with a small addition of carbon structures of different morphologies are considered while carbon–matrix composites with metal nanoparticles are poorly studied. Therefore, it is urgent to develop a technology for obtaining a similar type of composites, to study their mechanical properties, and to determine the mechanisms of deformation. In this regard, in the present work, a composite material consisting of crumpled graphene flakes filled with metallic nickel nanoparticles of two sizes is investigated. One of the promising methods for the fabrication of metal/graphene composites is proposed: hydrostatic compression at 0 K followed by annealing at two temperatures (1000 K and 2000 K).

## 2. Materials and Methods

In Figure 1, the initial structure of crumpled graphene with Ni nanoparticles of different diameters, $d_{Ni_{21}}$ = 5.5 Å (a) and $d_{Ni_{47}}$ = 7.2 Å (b), is shown. The size of nickel nanoparticles was chosen because a rigid graphene flake interacting with given size nanoclusters destroys their crystal structure, which does not happen with larger nanoparticles. The dynamics of the interaction of a graphene flake with nickel nanoclusters of different sizes was described in detail in our early work [52]. Taking into account the peculiarity of the interaction of crumpled graphene with nickel nanoparticles, it is possible to achieve a denser structure of the metal/graphene composite with a smaller pore volume fraction.

The graphene flakes (GFs) initially were armchair carbon nanotubes (11,11), cut along the *z*-axis to obtain a small flake. Graphene flakes filled with Ni nanoparticle were randomly rotated and translated 4×4×4 times along the *x*-, *y*-, and *z*-axes correspondingly to obtain 3D initial structure shown in Figure 1c,d. For simplicity, crumpled graphene was filled with Ni${}_{21}$ labeled as CG21 and with Ni${}_{47}$ labeled as CG47. The total number of atoms for CG21 was 17,472, where $N_{C}$ = 16 128 and $N_{Ni}$ = 1344, and for CG47 was 19,136, where $N_{C}$ = 16,128 and $N_{Ni}$ = 3008. To avoid overlap, the graphene flakes were placed far from each other in the initial structure. However, these pores quickly disappeared during compression. It should be mentioned that an increase in the size of the computation cell by two times did not lead to significant changes in the results [39,40].

Periodic boundary conditions were applied in all directions. All of the simulations were conducted using the LAMMPS package with the AIREBO [53] interatomic potential for the description of the interaction between carbon atoms, which included both covalent bonds in the basal plane of graphene flake and van der Waals interactions between GFs. For Ni–Ni and Ni–C interactions, simple pair Morse interatomic potential was used with the parameters for Ni–Ni [54] and for Ni–C [55,56]. The simulation configurations were visualized by Visual Molecular Dynamics (VMD) Software [57].

Equations of motion for the atoms were integrated numerically using the fourth-order Verlet method with the time step of 0.1 fs. A Nose-Hoover thermostat was used to control the system temperature under the NVT and NPT canonical ensembles (the substance amount (N), volume (V) or pressure (P) and temperature (T) are constant) when simulating the compression/tension of the structure and the annealing process, respectively.

It was shown earlier by the authors of [39,40] that it is impossible to obtain a composite material by hydrostatic compression at 0 K. Additionally, to obtain Ni/graphene composites, it was proposed to use high-temperature annealing after hydrostatic compression. The proposed technology for obtaining metal/graphene composites is one of the promising and simple methods for processing materials, which can be easily realized experimentally. In the experiment [10], a technology was proposed for the formation of a composite from Ni/graphene powder using compression followed by sintering at 1723 K. Thus, in the present work, the structures are sintered at high temperatures for 20 ps after hydrostatic compression. The temperature range from 1000 K to 2000 K was chosen so that the temperature would activate the formation of new chemical bonds between adjacent graphene flakes but would not allow for melting of the components. Note that the melting temperature of graphene is about 5000 K [58,59,60] and the melting point of the Ni nanocluster is 1728 K [10].

Before annealing, the structure was subjected to strain-controlled hydrostatic compression ($\varepsilon_{xx}=\varepsilon_{yy}=\varepsilon_{zz}=\varepsilon$) at the temperature close to 0 K with the strain rate $\dot{\varepsilon }$ = 0.01 ps${}^{-1}$. This is necessary to create a compact initial structure with a lower pore volume fraction. To study the mechanical properties of the obtained Ni/graphene composite, a tensile strain at 0 K was applied: hydrostatic tension and uniaxial tension along the *x*-, *y*-, and *z*-axes. The tension strain rate for each loading type was $\dot{\varepsilon }$ = 0.005 ps${}^{-1}$. The ultimate tensile strength ($\sigma_{UTS}$), which is the maximum that the material can achieve before breaking is considered one of the important characteristics for the composites under consideration. For hydrostatic tension, the value of hydrostatic pressure was calculated as *p*=($\sigma_{xx}+\sigma_{yy}+\sigma_{zz}$)/3.

## 3. Results

Figure 2a,c shows the structure of CG21 and CG47 after hydrostatic compression. It can be seen that, after deformation, a sufficient number of pores remain in the structure of crumpled graphene and that they periodically are distributed over the structure. This means that the formation of strong chemical bonds between GFs as a result of deformation did not occur. Moreover, the bigger the nanoparticle, the larger the pore size (see Figure 2c). The existence of pores in the Ni/graphene composite can have a negative effect on its strength properties, which is discussed below.

To reduce the pore size and to achieve the formation of new covalent bonds between graphene flakes, high-temperature annealing at 1000 K and 2000 K was used. After annealing at 1000 K, the number and size of pores in the compressed structure decreased (see Figure 2b,d). In the annealed CG21 structure (see Figure 2b), pores are almost not observed, and in the CG47 structure (see Figure 2d), after annealing at 1000 K, pores can be seen, although their size and volume fraction are small. An increase in the annealing temperature to 2000 K leads to a decrease in the volume fraction of pores in the CG47 structure.

Analysis of the composite structure before deformation showed that, during the initial stages of compression, structural elements (GFs with nanoparticle inside) rotated so that the zigzag edges of the graphene flake extends along the *x*-axis while the armchair edge extends along the *y*-axis, which can affect the deformation behavior. Therefore, to understand the effect of the initial orientation of the structural units on deformation behavior and mechanical properties, three directions of uniaxial tension are considered: along the *x*-, *y*-, and *z*-axes. Additionally, the hydrostatic tension of the obtained composites is estimated at which the effect of the initial orientation of GFs is negligible.

### 3.1. Hydrostatic Tension

In Figure 3a, the pressure–strain curves during hydrostatics tension for Ni/graphene composites obtained by hydrostatic compression at 0 K followed by annealing at 1000 K (solid line) and 2000 K (dashed line) are presented. It can be seen that the increase in the annealing temperature leads to an increase in the ultimate tensile strength ($\sigma_{UTS}$). After annealing at 2000 K, the $\sigma_{UTS}$ of CG21 (CG47) is 1.6 (1.5) times higher than at 1000 K. This can be explained by the fact that the annealing temperature 1000 K is not enough to melt a Ni particle [61] and not enough to mix carbon atoms and to form chemical bonds between carbon atoms. At the same time, such a temperature is sufficient to allow the spreading of Ni atoms inside the CG21 structure (see Figure 3b **I**).

Figure 3b shows that, after ultimate tensile strain is achieved (**I**, **II**, **III**, and **IV** points in Figure 3a), pores have appeared between GFs of crumpled graphene, which results in a stress decrease and rapture of the composite. At an annealing temperature of 1000 K, the $\sigma_{UTS}$ of the CG47 structure is 5.5% larger than that of the CG21, but the pores in the structure with Ni${}_{47}$ (CG47) appeared faster than for CG21. This can be explained by the fact that Ni${}_{47}$ nanoparticles almost completely fill the graphene nanoflake, thus limiting the ability of a graphene flake to form bonds between closely spaced carbon flakes, and those bonds that nevertheless formed between the flakes during annealing break quite easily in the process of hydrostatic tension.

Annealing at 2000 K does not lead to a significant difference in the stress–strain state of the CG21 and CG47 composites: $\sigma_{UTS}$ of CG21 and CG47 are very close (48.2 GPa and 47.6 GPa, respectively).

The process of fracturing the Ni/graphene composite after the ultimate tensile strength is achieved is shown on the example of GF filled with Ni${}_{21}$ and located in the center of the CG21 composite during hydrostatic tension after annealing at 2000 K (Figure 4a–d). Carbon atoms on the sides of the bonds broken during deformation are shown in blue. At the first stage (until the ultimate tensile strength is achieved), deformation occurs due to the unfolding of GFs, flattening of folds, and the formation of a more equilibrium structure. When the critical stress $\sigma_{UTS}$ is reached at ε = 0.23, the irreversible fracture of CG begins: pores and voids appear between graphene nanoflakes. However, from Figure 4c, it can be seen that, at ε = 0.24, the C–C interatomic bond (between blue atoms) is destroyed, which leads to the appearance of defect followed by pore formation. Therefore, failure of the composite occurs not only due to the formation of pores between neighboring GFs but also due to the rupture of interatomic covalent bonds in the structure of the nanoflakes. Since the breaking of these covalent bonds occurs gradually, the rapture of crumpled graphene does not proceed fast.

From Figure 4b–d, it can be seen that, for CG21 in the course of tension, Ni nanoparticles are divided into individual atoms. They settle on the inner surface of the graphene nanoflake and, as far as possible, occupy an equilibrium position above the center of the hexagons. Upon hydrostatic tension of the CG47 (see Figure 4e–g), the Ni${}_{47}$ nanoparticle maintains its spherical shape, although it becomes soft and amorphous after annealing. Note that a graphene nanoflake in CG47 fully covers the Ni${}_{47}$ nanoparticle, forming a strong unitary structural element, which slightly deforms during hydrostatic tension. Thus, the failure of such a composite under hydrostatic tension occurs only due to the formation of pores and voids graphene flakes, and the bonds inside the graphene flake are not damaged.

Below, the uniaxial tension of Ni/graphene composites obtained by hydrostatic compression with subsequent high-temperature annealing is considered. In Figures 5, 7 and 9, the stress–strain curves of the Ni/graphene composites under uniaxial tension along the *x*-, *y*- and *z*-axes correspondingly after annealing at 1000 K (solid lines) and 2000 K (dashed lines) are presented together with the characteristic snapshots of the structure under tension. Only the main stress component (labelled as σ) is presented for each loading direction: $\sigma_{xx}$ for tension along the *x*-axis, $\sigma_{yy}$ for tension along the *y*-axis, and $\sigma_{zz}$ for tension along *z*-axis. For detailed analysis of the structural transformations, tension of the single graphene flake filled with Ni${}_{47}$ nanoparticle along the *x*-, *y*-, and *z*-axes is presented in Figures 6, 8 and 10, respectively.

### 3.2. Uniaxial Tension along the *x*-Axis

Figure 5 shows the stress–strain curves of Ni/graphene composites CG21 and CG47 during uniaxial tension along the *x*-axis and the corresponding snapshots at the ultimate stress points. The tensile stress for CG21 annealed at 2000 K is about 25% higher than that after annealing at 1000 K. It is found that the structure of CG21 after annealing at 1000 K contains a larger number of pores and voids than that after annealing at 2000 K.

For the same reason, a difference of about 18% is observed in tensile stresses upon deformation of the CG47 composite after annealing at 1000 K and 2000 K at points **C** and **D** (red curves). However, tensile stresses for CG47 after annealing at both temperatures have grown to the value of 120 GPa (points **III** and **IV** on the red curves).

Structural analysis showed that, during uniaxial tension along the *x*-axis, a gradual rearrangement of the graphene structure occurs due to the breaking of old bonds and the formation of new bonds between carbon atoms. As a result, the volume fraction of pores and voids remaining in the Ni/graphene composite CG47 after annealing at 1000 K significantly decreased and the structure of this composite under tension became almost identical to the structure obtained after annealing at 2000 K (see Figure 5b **III** and **IV**). This can explain the equality of stresses at points **III** and **IV** on the red curves. In this case, Ni atoms are evenly distributed throughout the CG47 composite. Comparatively, for the CG21 composite, Ni atoms are distributed inhomogeneously over the graphene surface during tension (see Figure 5b **A** and **B**), which inhibits the decrease in the volume fraction of pores due to the transformation of the graphene structure.

Figure 6a–f shows the structure of one graphene flake of the CG47 composite annealed at 2000 K during uniaxial tension along the *x*-axis. Atoms in blue show carbon atoms located along the zigzag direction of GF. Other graphene flakes after hydrostatic compression and annealing are oriented along the *x*-axis in almost the same way as in Figure 6. Tension along the *x*-axis of the entire composite corresponds to the zigzag direction of graphene flake edge, which can lead to significant elongation of the obtained Ni/graphene structures to a strain of 3.5. As can be seen, bonds are straightened along *x*-axis before ε < 0.5–0.6 (see bonds between blue carbon atoms in Figure 6a–c). Then, at ε > 0.5–0.6, breaks of old covalent bonds and the formation of new bonds are observed (see Figure 6d–f). In this case, the hexagonal carbon rings are rearranged into other configurations, forming a complex defect structure. At high strains (ε > (1.5–1.6)), long carbon mono-chains are formed (Figure 6f) and amorphization of the graphene structure took place. Such monoatomic carbon chains are formed under large tensile strains, just before the fracture [62,63]. In Figure 6f, single carbon atoms belong to the neighboring GFs, which are not shown.

### 3.3. Uniaxial Tension along the *y*-Axis

Figure 7 shows the stress–strain curves of Ni/graphene composites CG21 and CG47 during uniaxial tension along the *y*-axis and the corresponding snapshots of the structure at critical points. As for hydrostatic tension, an increase in the annealing temperature to 2000 K leads to an increase in the ultimate tensile strength for CG21 and CG47 by 15% and 23%, respectively (see Figure 7a). A greater number of pores is observed under tension in the composite after annealing at 1000 K than after annealing at 2000 K. The formation of a large number of pores results in faster rapture of the composite and a decrease in the ultimate tensile strength. The number of pores for the CG21 composite is greater than that for CG47 (see Figure 7b); this can explain the difference in the elongation of Ni/graphene composites when $\sigma_{UTS}$ is achieved. The elongation of the composite CG21 after annealing at 1000 K (2000 K) is 128% (121%), and that for CG47 is 136% (132%). After reaching the $\sigma_{UTS}$ (points **I**, **II**, **III**, and **IV**), the stress/strain curves show a slow stress decrease.

During uniaxial tension of the Ni/graphene composites to the strain at points **A**, **B**, **C**, and **D** in Figure 7a, the structure is stretched without the formation of pores between the graphene flakes. From Figure 8a, it can be seen that GF is oriented by the armchair edge along the *y*-axis (along the loading direction). Tension of the composite results in a break in the carbon interatomic bonds (see Figure 8), the appearance of new bonds, and the formation of long carbon chains. Carbon atoms in these chains are separated by alternating single and triple bonds, which corresponds to a polyynic configuration [62,63]. Such a bond transformation is irreversible and leads to a sharp change in the stress–strain curves (see Figure 7a). At ε > 0.5, the formation of new structural elements composed of carbon atoms is observed, which differ from the graphene hexagonal structure (Figure 8c–e). At ε=1.72, the carbon structure is close to the amorphous state (see Figure 8f). Such a carbon structure transformation upon tension results in a strong elongation of the Ni/graphene composite by more than 170%. The Ni${}_{47}$ nanoparticle is destroyed during uniaxial tension and Ni atoms are uniformly distributed over the graphene surface. A similar transformation is observed for the Ni${}_{21}$.

### 3.4. Uniaxial Tension along the *z*-Axis

Figure 9 shows the stress–strain curves of Ni/graphene composites CG21 and CG47 during uniaxial tension along the *z*-axis and the corresponding snapshots of the structure at critical points. Under uniaxial tension along the *z*-axis, the stress–strain curves in Figure 9a do not qualitatively different from the curves shown in Figure 7a (tension along *x*-axis). However, an increase in the annealing temperature to 2000 K leads to a significant increase in the ultimate tensile strength. The tensile strength for the CG21 (CG47) composite annealed at 2000 K is 72% (54%) more than that of the composite annealed at 1000 K. Such a large difference in tensile stress is associated with a decrease in the bulk porosity of composites with an increase of the annealing temperature.

Figure 10a–f shows the central graphene flake of the CG47 composite after annealing at 2000 K. As in the case of tension along the *y* axis, at ε<0.5, a stretching of the carbon flake is observed and the following deformation (ε>0.5) results in the breaking of old bonds and the formation of a new one. At ε>1.1, a partial or complete transformation of the crumpled graphene structure into an amorphous state with the formation of long mono-atomic carbon chains is found (see Figure 10a–f).

## 4. Discussion

For correlation analysis, Figure 11 shows stress–strain curves for all of the tensile deformation schemes of the CG21 (Figure 11a) and CG47 (Figure 11b) composites annealed at 2000 K. This annealing temperature of Ni/graphene composites was chosen because it provides better formation of the homogeneous structure with the smallest volume fraction of pores and voids.

In Figure 11, it can be seen that hydrostatic tension (green thick line) to fracture of Ni/graphene composites occurs at the lowest stresses not exceeding 50 GPa and the strain at about 0.25. Such low values are associated with an intense increase in the volume fraction of pores and voids between GFs during hydrostatic tension. In this case, no transformation of the crumpled graphene structure into an amorphous state is observed.

The highest tensile stress value is achieved during uniaxial tension along the *x* axis, and this direction coincides with the zigzag edge of the GF. It is known that the graphene strength along the zigzag direction is higher than that along armchair direction. Thus, the stress–strain curve for tension along the *y*-axis is lower for both composites. In the case of tension along the *z*-direction, a single flake is oriented by its open side. During compression, two edges of the initial flake can interconnect between each other and form something similar to a small nanotube. The fracture takes place along the armchair-type edge of this new structural element. However, it is the weakest direction since the edges around open side are not connected by covalent bonds and can be easily opened.

Composite CG21 exhibits better compression since the Ni${}_{21}$ nanoparticle has a small size and can be deformed easier. Ni atoms spread inside the carbon structure and cannot be considered nanoparticles, just as separate atoms. Graphene flakes form new covalent bonds easier than in the case of the bigger nanoparticle. At high temperatures, Ni atoms can move inside the carbon network, and the final composite structure became more homogeneous.

In the structure with Ni${}_{47}$, graphene envelops the Ni nanocluster forming a rigid structural element; therefore, it is less deformed during hydrostatic compression [39,40,61]. Moreover, the Ni${}_{47}$ nanoparticle preserves a near-spherical shape even at high pressure, which leads to less anisotropy in stress–strain curves under uniaxial tension. Bigger nanoparticles prevent interactions between neighboring GFs, since part of the σ bond involves the van der Waals interaction between graphene flake and metal nanoparticle.

For all of the cases of uniaxial tension, structural amorphization occurred at ε>1.0. Such a structural transformation occurs due to the breaking of old carbon bonds and the formation of new bonds. The combination of these deformation processes leads to an increase in the ultimate strength and to a considerable elongation of the composites during uniaxial tension.

The observed difference in the stress–strain curves in Figure 11 is caused by the structural heterogeneity of the obtained CG21 and CG47 composites after hydrostatic deformation with the following annealing.

## 5. Conclusions

Molecular dynamic simulation is used to fabricate Ni/graphene composites with Ni${}_{21}$ and Ni${}_{47}$ nanoparticles by pressure-heat treatment. To obtain a composite, hydrostatic compression at 0 K followed by annealing at 1000 K and 2000 K is used. The strength of the obtained composites is evaluated using hydrostatic and uniaxial tension along the *x*-, *y*-, and *z*-axes.

It is found that all investigated composites have anisotropic strength properties. This is due to the fact that the volume fraction of pores and voids in the *x*, *y*, and *z* directions are very different. The most homogeneous structure is obtained for the CG47 composite after hydrostatic compression followed by annealing at 2000 K.

The best stress–strain behavior is observed under uniaxial tension along the *x*-axis of composites annealed at 2000 K. In the process of uniaxial tension, the breaking of old and the formation of new covalent bonds took place, which led to the transformation of the graphene structure into an amorphous one. Thus, a high strain is achieved before the failure of the composites. At large strain, long mono-atomic carbon chains are formed with alternating double and triple bonds between carbon atoms.

It is found that, using hydrostatic compression followed by annealing at 2000 K, it is possible to fabricate a Ni/graphene composite; however, it is necessary to achieve a more homogeneous structure with a smaller volume fraction of pores and voids, which will lead to a decrease or complete disappearance in strength property anisotropy.

## Figures and Tables

**Figure 1 materials-14-03087-f001:**
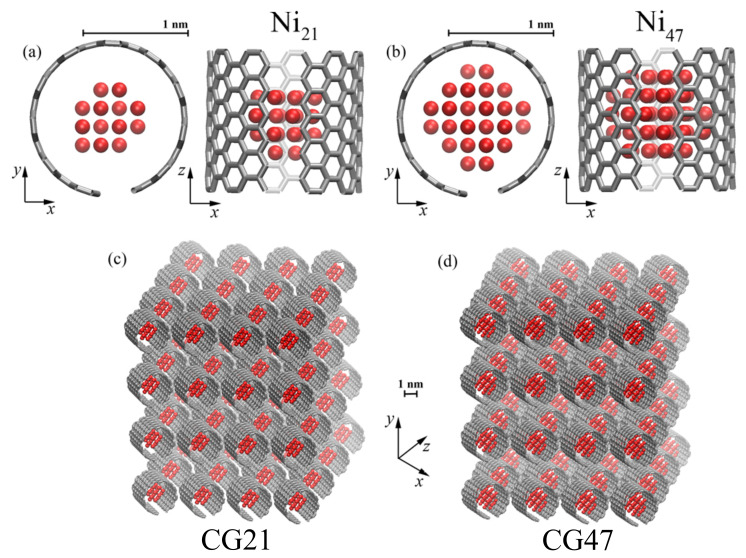
Initial structural unit—graphene flake filled with Ni nanoparticle Ni${}_{21}$ (**a**) and Ni${}_{47}$ (**b**) as the projection on xy and zx. (**c**,**d**) Structure of crumpled graphene filled with Ni${}_{21}$—CG21 (**c**) and Ni${}_{47}$—CG47 (**d**). **Ni** atoms are shown by red and **C** atoms—by grey color.

**Figure 2 materials-14-03087-f002:**
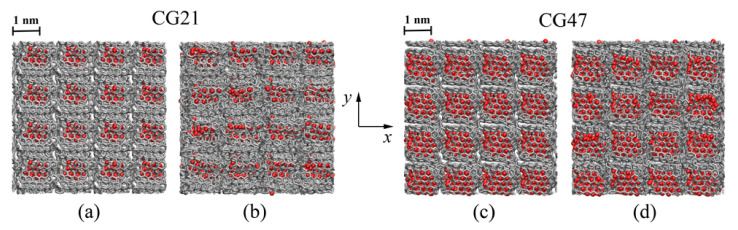
Snapshots of CG21 (**a**,**b**) and CG47 (**c**,**d**) before (**a**,**c**) and after (**b**,**d**) annealing at 1000 K. Colors as in Figure 1.

**Figure 3 materials-14-03087-f003:**
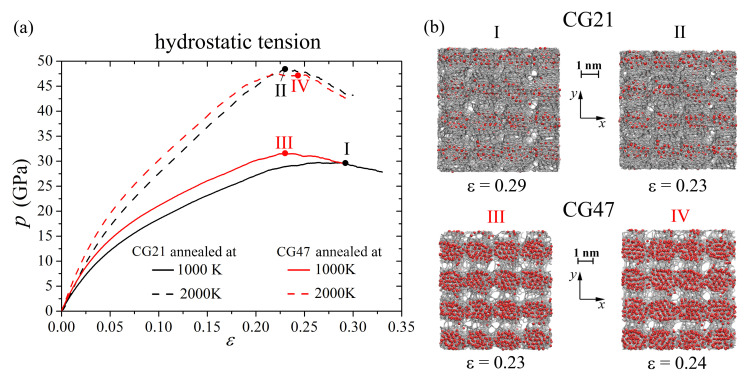
(**a**) Stress-strain curves during hydrostatics tension for Ni/graphene composite CG21 (black curves) and CG47 (red curves) after annealing at 1000 K (solid lines) and 2000 K (dashed lines). (**b**) Snapshots of CG21 (**I**—after annealing at 1000 K, **II**—after annealing at 2000 K) and CG47 (**III**—after annealing at 1000 K, **IV**—after annealing at 2000 K). Colors as in Figure 1.

**Figure 4 materials-14-03087-f004:**
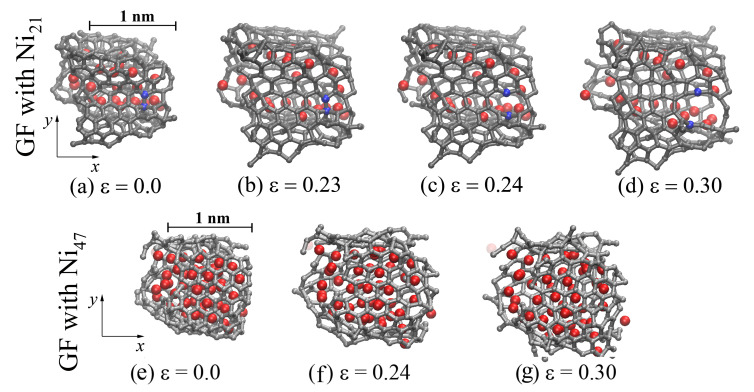
Snapshots of one GF (projection on the xy) during hydrostatic tension after annealing at 2000 K for CG21 (**a**–**d**) at ε = {0.0; 0.23; 0.24; 0.3} and CG47 (**e**–**g**) at ε = {0.0; 0.24; 0.3}. Colors as in Figure 1, but two carbon atoms are shown in blue to illustrate the breaking of bonds during deformation.

**Figure 5 materials-14-03087-f005:**
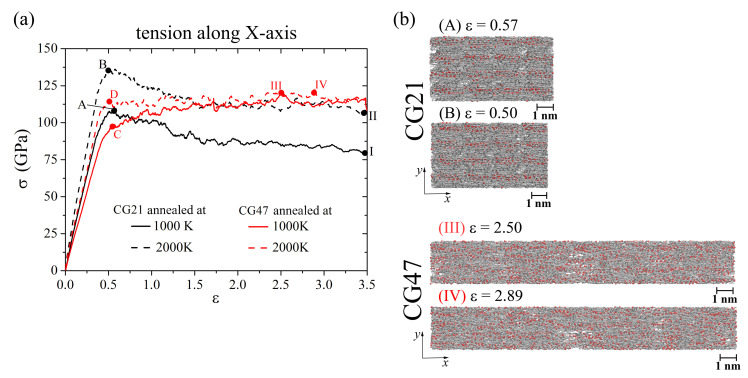
(**a**) Stress-strain curves during uniaxial tension along *x*-axis for CG21 (black curves) and CG47 (red curves) after annealing at 1000 K (solid line) and 2000 K (dashed line). (**b**) Snapshots of CG21 (**A,B**) and CG47 (**III**,**IV**) composites. Colors as in Figure 1.

**Figure 6 materials-14-03087-f006:**
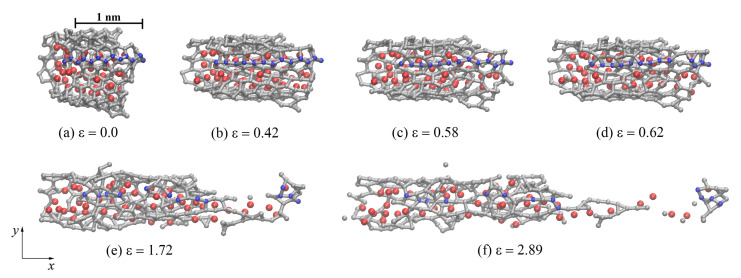
Snapshots of one GF (projection on the xy) for CG47 composite during uniaxial tension along the *x*-axis after annealing at 2000 K. Colors as in Figure 1, but a few carbon atoms are shown in blue to illustrate the direction of the zigzag graphene flake.

**Figure 7 materials-14-03087-f007:**
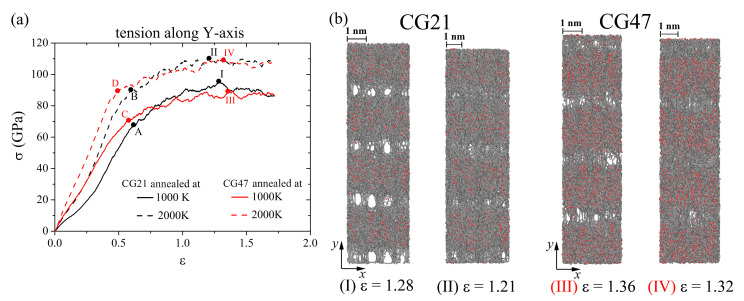
(**a**) Stress-strain curves during uniaxial tension along *y*-axis for CG21 (black curves) and CG47 (red curves) after annealing at 1000 K (solid line) and 2000 K (dashed line). (**b**) Snapshots of CG21 (**I**,**II**) and CG47 (**III**,**IV**) at the ultimate tensile strength. Colors as in Figure 1.

**Figure 8 materials-14-03087-f008:**
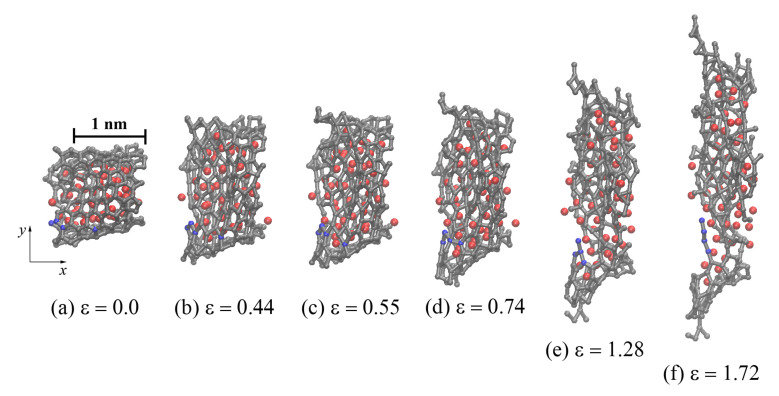
Snapshots of one GF (projection on the xy) for CG47 during uniaxial tension along the *y*-axis after annealing at 2000 K. Colors as in Figure 1, but a few carbon atoms are shown in blue to illustrate the breaking of bonds between them during deformation.

**Figure 9 materials-14-03087-f009:**
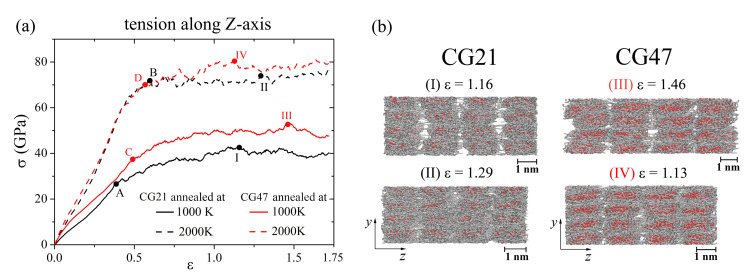
(**a**) Stress-strain curves during uniaxial tension along *z*-axis for CG21 (black curves) and CG47 (red curves) after annealing at 1000 K (solid line) and 2000 K (dashed line). (**b**) Snapshots of CG21 (**I**,**II**) and CG47 (**III**,**IV**). Colors as in Figure 1.

**Figure 10 materials-14-03087-f010:**
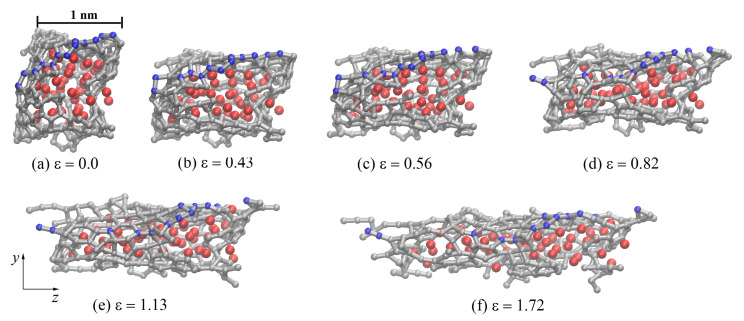
Snapshots of one GF (projection on the zy) for CG47 composite during uniaxial tension along the *z*-axis after annealing at 2000 K. Colors as in Figure 1, but a few carbon atoms are shown in blue to illustrate the breaking of bonds between them during deformation.

**Figure 11 materials-14-03087-f011:**
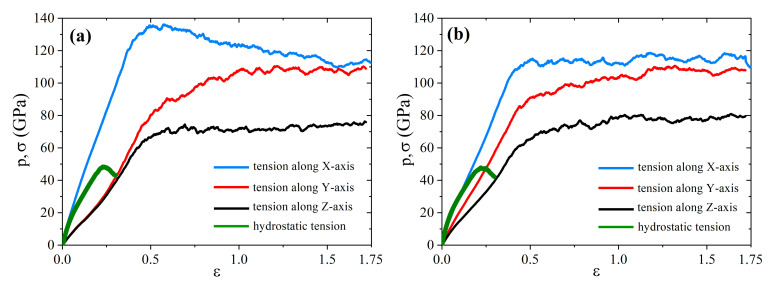
Stress–strain curves for Ni/graphene composite CG21 (**a**) and CG47 (**b**) after annealing at 2000 K. Here, σ is for uniaxial tension along *x*, *y* and *z* and *p* is for hydrostatic tension.

## Data Availability

Not applicable.

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
