# Peer review of "Effect of Nanoparticle Size on the Mechanical Strength of Ni–Graphene Composites"

_materials, 2021, doi:10.3390/ma14113087_

Round 1
Reviewer 1 Report
The paper “Effect of nanoparticle size on the mechanical strength of the Ni-graphene composite” researched the effect of nanoparticle size and annealing temperature on the Ni-graphene composite structure and mechanical strength. On the whole, the work is lack of novelty, the co-author has published similar work previously (the reference 20). The logic of the introduction is confusing simultaneously, so we can't see the necessity of this work. And the writing is not good to make the reading of the current manuscript tough. In addition, only two sizes of Ni were selected to explain the influence of nanoparticles size on the strength anisotropy of the composite, leading to the experiment slightly thin. More importantly, the figures should be explained more logical and combined with related mechanism.
Author Response
Comment 1
On the whole, the work is lack of novelty, the co-author has published similar work previously (the reference 20).
Reply:
We appreciate the comment. This manuscript is undoubtedly a continuation of the previously published (reference 20). It investigates different sizes of nickel nanoparticles (with 21 and 47 atoms) placed in a carbon structure, while in work [20] only the structure with Ni47 nanoparticles was considered. The difference in the works also lies in the fact that the presented work considers in sufficient detail the mechanism of deformation during the tension of the obtained nickel/graphene composites depending on the nickel nanoparticle size, which is of great scientific and practical interest. Corresponding explanations are added to the text.
Comment 2
The logic of the introduction is confusing simultaneously, so we can't see the necessity of this work.
Reply:
We appreciate the comment and thank the referee for this advice. The introduction has been significantly revised.
Comment 3
And the writing is not good to make the reading of the current manuscript tough.
Reply:
We appreciate the comment and thank the referee for this advice. Corresponding explanations are added to the text.
Comment 4
In addition, only two sizes of Ni were selected to explain the influence of nanoparticles size on the strength anisotropy of the composite, leading to the experiment slightly thin.
Reply:
We appreciate the comment. The reviewer is totally right; in the presented manuscript, the choice of the size of nickel nanoparticles was not explained in sufficient detail. The fact is that at the time of sending this article, a work devoted to the study of the interaction dynamics of a crumpled graphene flake with a nickel nanoparticle of various sizes from 21 to 78 atoms has not yet been published [Safina, L.R.; Krylova, K.A.; Murzaev, R.T.; Baimova, J.A.; Mulyukov, R.R. // Materials2021,14, 2098.]. It was shown in this work that a graphene flake easily covers a nickel nanocluster, but the dynamics of the interaction strongly depend on the particle size. A hard graphene flake easily deforms a small nickel cluster (21 atoms), and simply covers a large nickel nanoparticle (66 and 78 atoms) without making any changes to the structure of the nanocluster. A nanoparticle containing 47 nickel atoms is an intermediate value and interacting with a hard graphene flake, it is partially destroyed, losing its crystal structure. Since the goal of our work was to fabrication of Ni/graphene composite, based on the previous work, it was concluded that crumpled graphene with a nanocluster size of 78 atoms (or 66 atoms) cannot form sufficiently covalent bonds with neighboring flakes, since it is rigidly bound to this nanoscale. In this regard, only two sizes of nickel nanoparticles were chosen for the study: 21 atoms and 47 atoms. Corresponding explanations are added to the text.
Comment 5
More importantly, the figures should be explained more logical and combined with related mechanism.
Reply:
We appreciate the comment and thank the referee for this advice. The figures were explained more logically, generalizing the tensile deformation behavior of composites after different high-temperature annealing. All corresponding explanations are added to the text.

Reviewer 2 Report
This manuscript deals with an interesting topic, although well studied in the literature, i.e. the optimization of the synthesis parameters in the fabrication of Ni-graphene composites. It consists in a theoretical study, carried out by means of a molecular dynamics simulation, aiming specifically to assess the effect of the size of nickel nanoparticles, for the composite realization. According to the Authors’ analysis the main factors that affect the composite fabrication are the nanoparticle size, the orientation of the structural units and the temperature of the fabrication process.
Although I appreciate the reliability of the theoretical methods and the probable basic correctness of the final results, nevertheless I find the paper to at fault, concerning the comparison with the previously developed knowledge about Ni-graphene composites. In particular I wish to urge the Authors to compare their results with respect to the existing literature, e.g. Zhang et al., Sci. Adv. 2019;5 and also Zhaodi Ren et al 2015 Nanotechnology 26 065706. A nice and up to date review can be found in Journal of Materials Research and Technology, Volume 9, Issue 3, May–June 2020, Pages 6808-6833. So basically, I ask the Authors to duly compare with the abovementioned publications, showing at the same time the originality and discussing the progress of their approach.
Moreover, the paper is affected by many errors of grammar and synthax, such as:
Graphene is a new two-dimensional (2D) structures -> Graphene is a new two-dimensional (2D) structure, page 1
In the one hand-> On the one hand, page 2
Since the structure peculiarities considerably dependent -> Since the structure peculiarities considerably depend, page 2
and so on. In addition, articles are omitted in many places or used erratically.
I recommend the Authors to have a native English speaking person help them to revise the language of the manuscript.
Author Response
Comment 1
Although I appreciate the reliability of the theoretical methods and the probable basic correctness of the final results, nevertheless I find the paper to at fault, concerning the comparison with the previously developed knowledge about Ni-graphene composites. In particular, I wish to urge the Authors to compare their results with respect to the existing literature, e.g. Zhang et al., Sci. Adv. 2019;5 and also Zhaodi Ren et al 2015 Nanotechnology 26 065706. A nice and up to date review can be found in Journal of Materials Research and Technology, Volume 9, Issue 3, May–June 2020, Pages 6808-6833. So basically, I ask the Authors to duly compare with the abovementioned publications, showing at the same time the originality and discussing the progress of their approach.
Reply:
We thank the referee for this valuable comment since literature review improvement considerably increases the readability of the manuscript and the soundness of the results.
In [Zhang et al., Sci. Adv. 2019; 5] investigated composite materials from Ni powders and graphene sheets. In a shear mixing and freeze-drying process, graphene closely wrapped Ni powders. Further, the obtained Ni/graphene powder was subjected to compression, followed by sintering at 1450 °C. This method of obtaining Ni/graphene composite is similar to that proposed in our work. However, in [Zhang et al., Sci. Adv. 2019; 5], a Ni/graphene composite material with a carbon content of about 20 at.% was investigated. In our work, the amount of carbon was 84.3 and 92.3 at.% for the structures CG21 and CG47, respectively. Therefore, in our work, a composite material based on graphene is studied, and in [Zhang et al., Sci. Adv. 2019; 5] based on a nickel matrix. Such a significant difference in the objects of study affects the properties of the obtained composites and the deformation mechanisms.
In [Zhaodi Ren et al 2015 Nanotechnology 26 065706] Ni/graphene composite was synthesized by electrodeposition on the surface of a nickel matrix, which leads to an improvement in its mechanical properties. Our work aimed to show the possibility of forming a bulk graphene/nickel composite with a carbon content of more than 80 at.%.
A large and fairly complete review [Journal of Materials Research and Technology, Volume 9, Issue 3, May – June 2020, Pages 6808-6833] considers a large number of experimental works on the preparation and study of graphene-reinforced composites with a metal matrix (including nickel matrix). The properties of these composites are presented and the factors influencing these properties are explained. For example, the formation of graphene agglomerates negatively affects the mechanical properties of composites, and the formation of a coherent graphene-matrix interface leads to an increase in the mechanical properties. It has also been shown that the type of production of the composite has a strong influence on its properties. However, in all the composite materials presented in this review, the basis is a metal matrix, and in the work presented here, the basis is a graphene structure, which leads to the formation of a completely new type of composite with its unique properties that differ from the properties of a metal/graphene composite with a metal matrix. In addition, the dynamics of the deformation behavior of these two types of composite materials were also different. Since in composites with a metal matrix, mechanisms characteristic of plastic deformation of metals prevails, and when composites based on a carbon matrix are stretched, the graphene structure itself is deformed into a kind of amorphous structure with the formation of long carbon chains. This was reflected and described in detail in the work presented by us.
All corresponding explanations are added to the text.
Comment 2
Moreover, the paper is affected by many errors of grammar and synthax, such as:
Graphene is a new two-dimensional (2D) structures -> Graphene is a new two-dimensional (2D) structure, page 1
In the one hand-> On the one hand, page 2
Since the structure peculiarities considerably dependent -> Since the structure peculiarities considerably depend, page 2
and so on. In addition, articles are omitted in many places or used erratically.
I recommend the Authors to have a native English speaking person help them to revise the language of the manuscript.
Reply:
We thank the referee for such a careful reading. All syntax and grammar errors have been corrected.

Reviewer 3 Report
Krilova et al submitted a manuscript on the effect of nanoparticle size on the mechanical strength of the Ni-graphene composite. This study is theoretical and based on molecular dynamics simulation. The theoretical composite is to be synthesised at 0 K followed by annealing at 1000 and 2000K.
The study is relevant and the conclusions are based on their results; however, it is difficult to see the advantages of using such a difficult synthesis process involving extremely low and high temperatures. I recommend that they present in a figure and/or table the mechanical properties of the composites that were produced experimentally, indicating the synthesis conditions.
Author Response
Comment 1
The study is relevant and the conclusions are based on their results; however, it is difficult to see the advantages of using such a difficult synthesis process involving extremely low and high temperatures. I recommend that they present in a figure and/or table the mechanical properties of the composites that were produced experimentally, indicating the synthesis conditions.
Reply:
We appreciate the comment and thank the referee for this advice. A table was added to the manuscript, which reflects the most common methods for obtaining Ni/graphene composites. Among the presented methods is close to the one proposed in our work. It is based on compression of a Ni/graphene powder mixture followed by sintering at 1723 K. Note that in our work, hydrostatic compression was carried out at zero temperature, which is difficult to achieve experimentally. However, thermal fluctuations of atoms are excluded, and the deformation process was used to reduce the volume fraction of pores and voids of the Ni/graphene structure. In the future, we plan to carry out hydrostatic compression of composites at room temperature, bringing our method of obtaining Ni/graphene composites closer to experimental ones. Corresponding explanations are added to the text.

Round 2
Reviewer 1 Report
All questions have been reasonable answered and modified.In addition, it is suggested to modify the format of the table.
Author Response
Thank you very much. The table was corrected according to the requirements of the journal MDPI Materials.
Reviewer 2 Report
The Authors improved the manuscript, following my recommendations. The paper is now publisheable in its present form.
Author Response
Thank you very much!
Reviewer 3 Report
The authors made the requested modifications. The manuscript can be accepted for publication.
Author Response
Thank you very much.